# Close Surgical Margins in Oral and Oropharyngeal Cancer: Do They Impact Prognosis?

**DOI:** 10.3390/cancers14122990

**Published:** 2022-06-17

**Authors:** Dong-Hyun Lee, Geun-Jeon Kim, Hyun-Bum Kim, Hyun-Il Shin, Choung-Soo Kim, Inn-Chul Nam, Jung-Hae Cho, Young-Hoon Joo, Kwang-Jae Cho, Dong-Il Sun, Young-Hak Park, Jun-Ook Park

**Affiliations:** Department of Otolaryngology-Head and Neck Surgery, Catholic University College of Medicine, Seoul 21431, Korea; cjidea@naver.com (D.-H.L.); emelanciana@naver.com (G.-J.K.); goldgold11@hanmail.net (H.-B.K.); marineshs@naver.com (H.-I.S.); drchoung@catholic.ac.kr (C.-S.K.); entnam@catholic.ac.kr (I.-C.N.); jhchomd@catholic.ac.kr (J.-H.C.); joodoct@catholic.ac.kr (Y.-H.J.); entckj@catholic.ac.kr (K.-J.C.); hnsdi@catholic.ac.kr (D.-I.S.); yhpark7@catholic.ac.kr (Y.-H.P.)

**Keywords:** oral cavity squamous cell carcinoma, oropharyngeal squamous cell carcinoma, main specimen margins, close margins, survival

## Abstract

**Simple Summary:**

The overall survival of patients with close margins was no different than that of others when appropriate postoperative adjuvant and/or salvage treatment were/was prescribed. However, we could not determine the impact of close margins on locoregional recurrence given various biases in our study setting.

**Abstract:**

Introduction. Mucosal margins exhibit a mean shrinkage of 30–40% after resection of oral and oropharyngeal cancers, and an adequate in situ surgical margin frequently results in a pathological close margin. However, the impact on prognosis remains unclear. We investigated the impact of a pathological close margin on disease-free survival (DFS) and overall survival (OS). Methods. We retrospectively reviewed the clinicopathological data of 418 patients diagnosed with squamous cell carcinomas of the oral cavity or oropharynx who underwent initial surgery (with curative intent) at our institute between 2010 and 2016. Results. Of the total population, the pathological marginal status of 290 (69.4%) patients was reported as clear (>5 mm), 61 (14.6%) as close (>1 mm, ≤5 mm), and 67 (16.0%) as positive (≤1 mm). The 5-year DFSs were 79.3%, 65.1%, and 52% in patients in the negative margin (group 1), close margin (group 2), and positive margin (group 3) groups, respectively. The difference between groups 1 and 2 was not significant (*p* = 0.213) but the difference between groups 2 and 3 was (*p* = 0.034). The 5-year OSs were 79.4%, 84%, and 52.3% in groups 1, 2, and 3, respectively. The difference between groups 1 and 2 was not significant (*p* = 0.824) but the difference between groups 2 and 3 was (*p* = 0.001). In multivariate analysis, older age, advanced T stage, and a positive margin were independently prognostic of the 5-year DFS and OS. Conclusion. In conclusion, the OS of patients with close margins was no different than that of others when appropriate postoperative adjuvant and/or salvage treatment were/was prescribed. However, we could not determine the impact of close margins on locoregional recurrence given various biases in our study setting. A future prospective study is needed.

## 1. Introduction

The current management of head-and-neck cancers features surgery alone in patients with early-stage disease lacking poorly prognostic factors but surgery combined with adjuvant treatment in patients with any poor prognostic factor including an insufficient surgical margin [1]. The guidelines of the Royal Society of Pathology define a pathological surgical margin of more than 5 mm as clear, a margin of 1–5 mm as close, and a margin less than 1 mm as positive [2]. The National Comprehensive Cancer Network (NCCN) guidelines describe a clear surgical margin as histological confirmation of a distance of at least 5 mm from the invasive tumor to the resected margin [3]. A positive surgical margin is a well-known, poor prognostic factor, and has been considered the prime risk factor associated with local recurrence and poor survival [4,5,6]. However, whether the prognosis of patients with close surgical margins is really poor remains controversial [7,8,9,10,11], as does whether all cases with close surgical margins should be prescribed adjuvant therapy [12].

In practice, surgeons frequently encounter ambiguous, pathological close surgical margins. Sometimes, it is impossible to ensure appropriate surgical margins given the structural characteristics of the maxilla, mandible, and posterior pharyngeal wall. In addition, surgeons are sometimes disappointed that the pathological results indicate insufficient margins; they are certain that the tumors were sufficiently (widely) resected. Mucosal surgical margins usually shrink after electrosurgery, formalin fixation, and slide preparation. Buccal mucosal margins shrink by more than 45% and tongue mucosal margins shrink by 30%, which means that a 7 mm resection in situ margin is frequently a close pathological margin (less than 5 mm) [13]. Therefore, to ensure a sufficient pathological margin, tumor resection that includes 1–1.5 cm of normal mucosa is required, given the expected shrinkage [3,13]. It may be good oncological practice to remove a cancer with a wide margin, but this may compromise the functional results and the quality of life [14].

Herein, we investigate the relationship between a close surgical margin and disease-free survival (DFS) and overall survival (OS).

## 2. Materials and Methods

### 2.1. Materials

We retrospectively reviewed the clinical and pathological data of patients diagnosed with squamous cell carcinomas of the oral cavity and oropharynx who underwent surgical treatment in our institute between 2010 and 2016. Our Institutional Review Board approved this retrospective review of medical records and the use of archived tumor specimens (approval no. XC19RCDI0096K). A total of 418 patients who met the following criteria were included: surgically resectable oral or oropharyngeal cancer (stages T1–4, N1–3, or M0); no previous treatment prior to hospitalization; initial surgery with curative intent regardless of neoadjuvant and/or adjuvant treatment; and the availability of pathological reports on surgical margin status. All patients were staged using the American Joint Committee on Cancer (AJCC) edition 8 system. We evaluated age, sex, primary tumor site, neo-adjuvant and/or adjuvant treatment, surgical data, clinical and pathological stages, perineural invasion and/or vascular invasion, P16 status, and the pathological surgical margin.

### 2.2. Treatment Modalities

Primary lesions of the oral cavity and oropharynx were resected via a transoral approach or using lateral pharyngotomy or mandibulotomy. Prophylactic (selective) neck dissection was performed in patients with clinically negative neck sites and modified radical neck dissection was employed to treat clinically positive neck sites. Our multidisciplinary team prescribed postoperative adjuvant treatment after considering the pathological results (surgical margin status; vascular, lymphatic, and perineural invasion status; the number and extracapsular extensions of lymph node metastases; and the general condition of the patient). Most patients were followed up every 6 months for the first 2 years and every 12 months thereafter via computed tomography (CT) or magnetic resonance imaging (MRI), and/or liver/bone scan and/or positron emission tomography (PET) CT.

### 2.3. Histopathological Review

Resected tissues were fixed in 10% (*v*/*v*) formaldehyde, embedded in paraffin wax, and evaluated by a pathologist who specialized in head-and-neck pathology. We focused on intraoperative margin sampling from the patient tumor bed and permanent margin sampling from the principal specimen. The margin was considered positive if invasive cancer was present at the inked edge or <1 mm from the edge, close if 1–5 mm from the edge, and negative if >5 mm from the edge [2].

### 2.4. Statistical Analyses

Student’s *t*-test was used to analyze continuous data that were normally distributed and the Mann–Whitney U-test was used to compare data with skewed distributions; categorical data were compared using the chi-square test. Five-year OS was the interval between the date of surgery and either the date of death (an event) or the last follow-up (censoring); OSs are reported as medians with 95% confidence intervals. Five-year DFS was the time between the date of surgery and the date of diagnosis of locoregional recurrence or distant metastasis. The overall time to recurrence was the interval between the date of surgery and the first recurrence (either locoregional or a distant metastasis). If a patient died without evidence of recurrence (censored), the date of last follow-up imaging or last follow-up without clinical signs of recurrent disease was used. Therefore, death was not a competing risk in analyses. Kaplan–Meier analysis and the log-rank test were used to evaluate differences in recurrence and survival between groups.

Characteristics associated with recurrence or survival were included in a Cox’s proportional hazards regression model. A *p*-value < 0.05 was deemed to indicate statistical significance. All analyses were performed using SPSS software (ver. 20.0; SPSS, Inc., Chicago, IL, USA).

## 3. Results

The study population consisted of 418 patients of mean age 58.5 years and the primary cancer sites were the oral cavity (*n* = 241) and oropharynx (*n* = 117). Patient characteristics, treatment modalities, and outcomes are summarized in Table 1. Of the total population, the pathological marginal status was clear in 290 (69.4%) patients, close in 61 (14.6%) patients, and positive in 67 (16.0%). The 5-year DFS did not significantly differ between patients with clear and close margins (*p* = 0.213) but did between patients with close and positive margins (*p* = 0.034). The 5-year OS did not significantly differ between patients with clear and close margins (*p* = 0.824) but did between patients with close and positive margins (*p* = 0.001) (Table 2). The Kaplan–Meier curves for survival outcomes by marginal status are shown in Figure 1. The 5-year DFSs were 79.3%, 65.1%, and 52% in patients with clear, close, and positive margins, respectively. The 5-year OSs were 79.4%, 84%, and 52.3% in patients with clear, close, and positive margins, respectively. In univariate analysis of DFS, older age (≥65 years), advanced TN stage, surgical margin, and perineural and vascular invasion status were significant. In univariate analysis for OS, older age (≥65 years), advanced TN stage, adjuvant therapy, surgical margin, perineural and vascular invasion status, and P16 status were significant (Table 3). In multivariate analysis, older age, advanced T stage, and a positive margin were predictive of DFS and OS, but a close margin was not (Table 4).

## 4. Discussion

The 5-year DFS and OS did not significantly differ between patients with clear and close margins but did between patients with close and positive margins. The recurrence rate in our study was higher (79.3% vs. 65.1%) in close margin patients than clear margin patients, although the difference was not significant. Mortality rates did not differ between the two groups. Collectively, our findings indicated that appropriate adjuvant therapies in patients with close margins produced similar survival rates to those of clear margin patients. Since the proportion of adjuvant therapy was higher (67% vs. 43%) in close margin patients, it is unclear whether close margins could be locally controlled by surgery alone. A randomized controlled trial is required to address this question.

Older age, advanced T-stage, and a positive surgical margin were independent poor prognostic factors. Although old age is a common risk factor for postoperative complication and poor survival outcome, chronological age alone should not be a contraindication to aggressive surgical approach, which should be attempted whenever risk-assessment ration is favorable. Surgery should be offered as the preferred treatment in consideration of potential functional outcome, the Comprehensive Geriatrics Assessment (CGA), the life expectancy and, importantly and the patient’s wishes [15,16,17]. The CGA is a standardized and validated tool that includes medical, functional, psychological, cognitive, and social well-being aspects [18]. For the T classification, the 8th AJCC emphasizes the negative effect of increasing depth of invasion in oral cancer. Pathologically, the depth of invasion is determined by drawing a line connecting the basement membrane of the nearest normal mucosa to the deepest point. Thus, staging is not determined by surface size alone. Recent studies have suggested that depth of invasion is more predictive than tumor thickness; penetration of the extrinsic muscle of the tongue at T4 is not used as a criterion [19,20].

Several studies have sought to identify minimal, surgical margin cutoffs that can afford an acceptable prognosis. Hinni et al. subjected 128 tonsil cancer patients to microscopic resection mapping. When the nearest deep surgical margin was 1.98 mm and the nearest mucosal margin was 2.82 mm, the 5-year local treatment rate was 99%, the DFS was 94.5%, and the OS was 76% [8]. Zanoni et al. reported that the DFSs were equivalent in oral cancer patients with pathological margins of 2.3–5 and greater than 5 mm [9]. Binahmed et al. compared the survival of oral cancer patients with pathological margins of 2 mm and over; the survival rates were similar [12]. Brickman et al. showed that a 3 mm pathological margin cutoff was significantly associated with better DFS and OS in patients with oral cavity cancers [11]. Tasche et al. reported that a pathological margin less than 1 mm in oral cancer patients was associated with an increased risk for local recurrence; such patients might benefit from additional treatment [10]. To ensure that the cutoff value is met, the frozen section margin should perhaps be added to the final pathological margin, but this is difficult given the variabilities in gross tumor clearance, where and how extra sections are taken, and how the sections are interpreted. A previous study demonstrates that positive fresh frozen margins, regardless of re-resection to clear margin, could be a powerful risk factor that determines a poor oncologic outcome [21]. Some recent articles suggest that preoperative ultrasound can accurately identify the tumor margins and contributes to the therapeutic plans [22,23].

The limitations of this study include its retrospective nature, the small number of patients, and the fact that it was a single-institution work. A multicenter, randomized, controlled clinical trial is required.

## 5. Conclusions

The OS of patients with close margins was no poorer than that of others when appropriate postoperative adjuvant and/or salvage treatment were/was prescribed. However, we could not determine the impact of close margins on locoregional recurrence given various biases in our study setting. A future prospective study is needed.

## Figures and Tables

**Figure 1 cancers-14-02990-f001:**
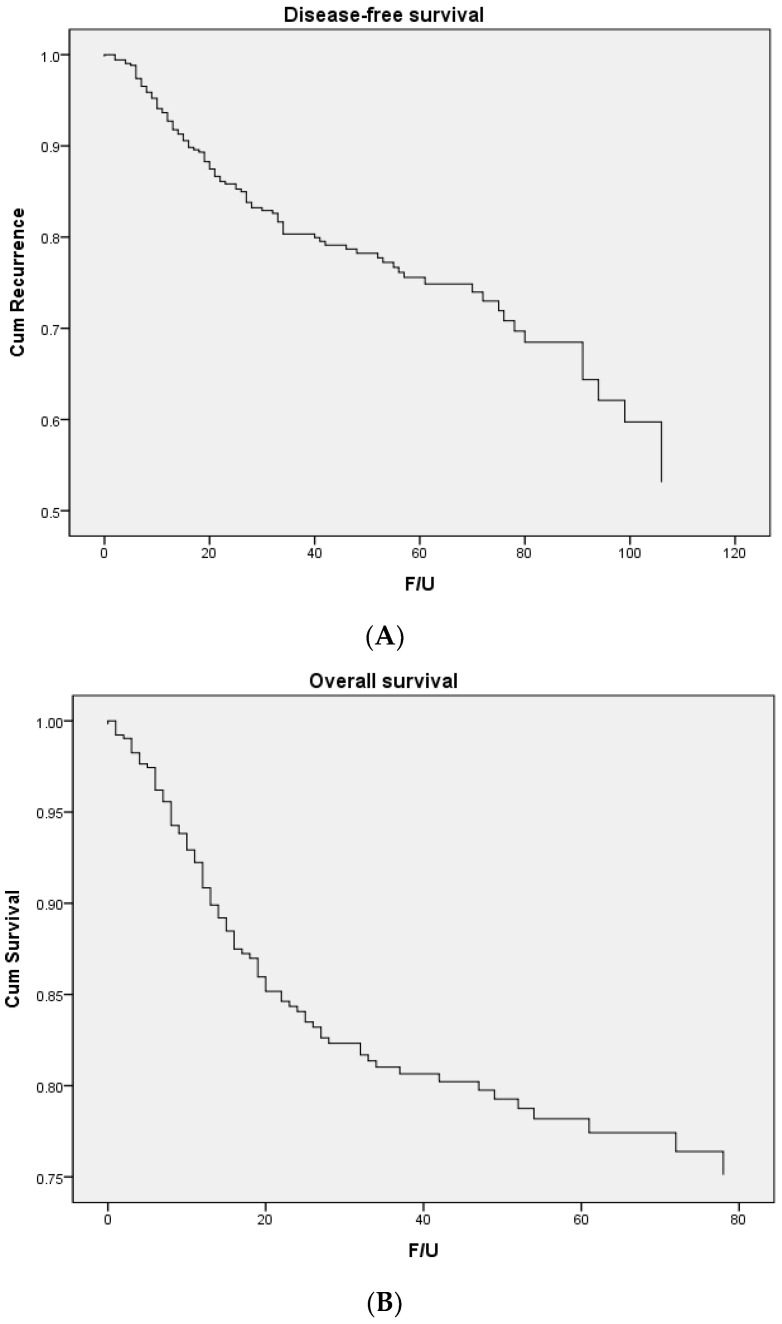
Kaplan–Meier curves of the survival outcomes of oral/oropharyngeal cancer patients (*n* = 418). (**A**) Disease-free survival, (**B**) overall survival, (**C**) disease-free survival by the surgical margin, and (**D**) overall survival by the surgical margin.

**Table 1 cancers-14-02990-t001:** Patient characteristics, treatment modalities, and outcomes (*n* = 418).

Parameter		Value (%)
Sex	Male	300 (71.8)
	Female	118 (28.2)
Age (years)		58.50 ± 13.40
Primary site	Oral cavity	241 (57.7)
	Oropharynx	177 (42.3)
AJCC stage	T1	156 (37.3)
	T2	162 (39.0)
	T3	59 (14.1)
	T4	40 (9.6)
	N0	220 (52.6)
	N1	74 (17.7)
	N2	117 (28)
	N3	7 (1.7)
Treatment modality	Surgery only	202 (48.3)
	Surgery + adjuvant therapy	216 (51.7)
Mean follow-up duration (months)		42 ± 28
Outcomes	5-year DFS	73.3%
	5-year OS	76.0%

Abbreviations: AJCC, American Joint Committee on Cancer; DFS, disease-free survival; OS, overall survival.

**Table 2 cancers-14-02990-t002:** Effect of surgical margin status on survival and recurrence after surgery to treat oral and oropharyngeal cancers (*n* = 418).

Pairwise Comparison			
5-Year DFS		5-Year OS	
Surgical Margin	*p*-Value	Surgical Margin	*p*-Value
Clear vs. close	0.213	Clear vs. close	0.824
Close vs. positive	0.034 *	Close vs. positive	0.001 *
Clear vs. positive	<0.001 *	Clear vs. positive	<0.001 *

* Significant at *p* < 0.05. Abbreviations: DFS, disease-free survival; OS, overall survival; Clear, pathological surgical margin > 5 mm; Close, pathological surgical margin ≤ 5 mm and >1 mm; Positive, pathological surgical margin ≤ 1 mm.

**Table 3 cancers-14-02990-t003:** Univariate analysis of parameters predicting the prognosis of oral and oropharyngeal cancer patients (*n* = 418).

Parameter		Number of Patients (%)	5-Year DFS (SD)	*p*-Value	5-Year OS (SD)	*p*-Value
Age (years)	<65	278 (66.5)	80.1 (2.7)	<0.001 *	90.5 (2.5)	<0.001 *
	≥65	140 (33.5)	59.9 (5.1)		62.9 (4.8)	
Primary site	Oral cavity	241 (57.7)	72.0 (3.3)	0.273	74.4 (3.2)	0.570
	Oropharynx	177 (42.3)	75.8 (3.7)		78.0 (3.4)	
T stage	T 1/2	319 (76.3)	80.5 (2.5)	<0.001 *	92.1 (2.4)	0.001 *
	T 3/4	99 (23.7)	52.2 (5.9)		67.5 (5.8)	
N stage	N 0/1	294 (70.3)	77.0 (2.8)	0.007 *	80.6 (2.6)	0.024 *
	N 2/3	124 (29.7)	66.6 (4.6)		74.6 (4.8)	
Treatment	Surgery only	202 (48.3)	77.1 (3.4)	0.129	83.0 (3.0)	0.005 *
	Surgery + adjuvant therapy	216 (51.7)	70.6 (3.5)		71.4 (3.5)	
Surgical margin	Clear	290 (69.4)	79.3 (3.4)	<0.001 *	79.4 (2.6)	0.004 *
	Close and positive	128 (30.6)	59.1 (4.8)		67.7 (4.7)	
	Clear and close	351(84.0)	76.7 (2.9)	<0.001 *	80.2 (2.4)	<0.001 *
	Positive	67 (16.0)	53.2 (6.1)		52.3 (6.0)	
PNI	(−)	311 (74.4)	77.5 (2.9)	0.009 *	80.1 (2.5)	0.001 *
	(+)	65 (15.6)	60.7 (6.2)		55.8 (6.5)	
	N/A	42(10.0)				
Lymphatic invasion	(−)	232 (55.5)	76.9 (2.9)	0.106	77.5 (2.6)	0.162
	(+)	147 (35.2)	69.2 (4.2)		73.4 (4.1)	
	N/A	39 (9.3)				
Vascular invasion	(−)	349 (83.5)	76.0 (2.7)	0.001 *	77.0 (2.4)	0.038 *
	(+)	25 (6.0)	51.6 (8.4)		61.2 (8.5)	
	N/A	44 (10.5)				
P16	(−)	183 (43.8)	74.6 (3.7)	0.231	71.5 (3.3)	0.030 *
	(+)	174 (41.6)	80.3 (3.4)		83.5 (3.1)	
	N/A	61 (14.6)				

* Significant at *p* < 0.05. Abbreviations: DFS, disease-free survival; SD, standard deviation; OS, overall survival; Clear, pathological surgical margin > 5 mm; Close, pathological surgical margin ≤ 5 mm and >1 mm; Positive, pathological surgical margin ≤ 1 mm; PNI, perineural invasion; N/A, not available.

**Table 4 cancers-14-02990-t004:** Multivariate analysis of recurrence and overall survival in patients with oral/oropharyngeal cancers (*n* = 418).

Parameter		5-Year DFS			5-Year OS		
		HR	95% CI	*p*-Value	HR	95% CI	*p*-Value
Older age	≥65	2.138	1.403–3.259	0.001 *	2.251	1.451–3.490	0.001 *
Advanced T stage	T3/4	2.717	1.391–5.309	0.003 *	1.672	0.979–2.858	0.050 *
Advanced N stage	N2/3	1.515	0.960–2.391	0.075	1.523	0.948–2.447	0.082
Adjuvant treatment					1.650	0.882–3.089	0.117
Surgical margin	Close	1.535	0.913–2.581	0.106	0.760	0.384–1.507	0.432
	Positive	1.763	1.030–3.017	0.039 *	1.826	1.076–3.098	0.026 *
PNI		1.568	0.899–2.736	0.113	1.734	0.950–3.167	0.073
Vascular invasion		1.575	0.779–3.184	0.206	0.886	0.340–2.305	0.803
P16					0.682	0.401–1.158	0.156

* Significant at *p* < 0.05. Abbreviations: DFS, disease-free survival; OS, overall survival; HR, hazard ratio; CI, confidence interval; Close, pathological surgical margin ≤ 5 mm and >1 mm; Positive, pathological surgical margin ≤ 1 mm; PNI, perineural invasion.

## Data Availability

Data available upon request due to data sharing restrictions. The data presented in this study are available upon request from the corresponding author.

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
