# Peer review of "Close Surgical Margins in Oral and Oropharyngeal Cancer: Do They Impact Prognosis?"

_cancers, 2022, doi:10.3390/cancers14122990_

Round 1

Reviewer 1 Report

The authors conducted a single institution, retrospective analysis of overall survival and disease-free survival of patients who received therapeutic surgery for squamous carcinomas of the oral cavity and/or oropharynx and found that overall survival was significantly poorer in those who had positive resection margins than in those who had clear or close margins. The manuscript is overall well written, and both the methods and results sections are clear and consistent. Anyway, the discussion section could be further expanded to increase the educational content of the manuscript, e.g., by adding recent research on the use of intraoral ultrasonography to study the resection margins of oral cancers; authors could be interested in the manuscripts by Limongelli et al. (Limongelli L, Capodiferro S, Tempesta A, et al. Early tongue carcinomas (clinical stage I and II): echo-guided three-dimensional diode laser mini-invasive surgery with evaluation of histological prognostic parameters. A study of 85 cases with prolonged follow-up. Lasers Med Sci. 2020;35(3):751-758. doi:10.1007/s10103-019-02932-z) and by Angelelli et al. (Angelelli G, Moschetta M, Limongelli L, et al. Endocavitary sonography of early oral cavity malignant tumors. Head Neck. 2017;39(7):1349-1356. doi:10.1002/hed.24779).

In my opinion, the current manuscript should be accepted after those minor revisions.

With warm regards

Author Response

Point : the discussion section could be further expanded to increase the educational content of the manuscript, e.g., by adding recent research on the use of intraoral ultrasonography to study the resection margins of oral cancers; authors could be interested in the manuscripts by Limongelli et al. (Limongelli L, Capodiferro S, Tempesta A, et al. Early tongue carcinomas (clinical stage I and II): echo-guided three-dimensional diode laser mini-invasive surgery with evaluation of histological prognostic parameters. A study of 85 cases with prolonged follow-up. Lasers Med Sci. 2020;35(3):751-758. doi:10.1007/s10103-019-02932-z) and by Angelelli et al. (Angelelli G, Moschetta M, Limongelli L, et al. Endocavitary sonography of early oral cavity malignant tumors. Head Neck. 2017;39(7):1349-1356. doi:10.1002/hed.24779).

Response 1: We have added this sentence in the discussion.

"Some recent articles suggest that preoperative ultrasound can accurately identify the tumor margins and contributes to the therapeutic plans [22,23]."

22. Angelelli, G.; Moschetta, M.; Limongelli, L.; Albergo, A.; Lacalendola, E.; Brindicci, F.; Favia, G.; Maiorano, E. Endocavitary sonography of early oral cavity malignant tumors. Head Neck 2017, 39, 1349-1356, doi:10.1002/hed.24779.

23. Limongelli, L.; Capodiferro, S.; Tempesta, A.; Sportelli, P.; Dell'Olio, F.; Angelelli, G.; Maiorano, E.; Favia, G. Early tongue carcinomas (clinical stage I and II): echo-guided three-dimensional diode laser mini-invasive surgery with evaluation of histological prognostic parameters. A study of 85 cases with prolonged follow-up. Lasers Med Sci 2020, 35, 751-758, doi:10.1007/s10103-019-02932-z.

We appreciate your positive feedback.

Sincerely yours

Reviewer 2 Report

Thank you for the opportunity to review this interesting article on the prognostic role of close surgical margins in oral and oropharyngeal cancer.

The introduction is comprehensive and not too long, the statistical methods are appropriate, the results are presented clearly and the discussion includes the main aspects and a limitations section.

The reference citations are sufficient and does not include inappropriate self-citations.

From my point of view there are only some minor issues to be corrected.

#List of authors: Please correct the last author: "and Jun-Ook Park, MD" instead of "Jun-Ook Park and MD".

#Table 1: Please provide a legend for abbreviations "AJCC", "DFS", and "OS".

#Tables 3 and 4: Please provide explanation for "P16".

#Table 4: 5-year OS: "1..451-3.490" please remove typo.

Thank you again.

Author Response

From my point of view there are only some minor issues to be corrected.

#List of authors: Please correct the last author: "and Jun-Ook Park, MD" instead of "Jun-Ook Park and MD".

#Table 1: Please provide a legend for abbreviations "AJCC", "DFS", and "OS".

#Tables 3 and 4: Please provide explanation for "P16".

#Table 4: 5-year OS: "1..451-3.490" please remove typo.

Response : Thankfully revised your delicate feedback.

But p16 (also known as p16INK4a, cyclin-dependent kinase inhibitor 2A) is a protein that slows cell division by slowing the progression of the cell, thereby acting as a tumor suppressor.

p16 is widely being used as a biomarker in the prevention of oropharyneal squamous cell carcinoma in this field.
So we don't think there's any need for further explanation.

Sincerely yours